

# Current operators in integrable spin chains: lessons from long range deformations

**Balázs Pozsgay**

MTA-BME Quantum Dynamics and Correlations Research Group,
Department of Theoretical Physics,
Budapest University of Technology and Economics, 1521 Budapest, Hungary

## Abstract

We consider the finite volume mean values of current operators in integrable spin chains with local interactions, and provide an alternative derivation of the exact result found recently by the author and two collaborators. We use a certain type of long range deformation of the local spin chains, which was discovered and explored earlier in the context of the AdS/CFT correspondence. This method is immediately applicable also to higher rank models: as a concrete example we derive the current mean values in the $SU(3)$-symmetric fundamental model, solvable by the nested Bethe Ansatz. The exact results take the same form as in the Heisenberg spin chains: they involve the one-particle eigenvalues of the conserved charges and the inverse of the Gaudin matrix.


# 1 Introduction

Recently there has been some interest in the computation of current mean values in one dimensional integrable models. The motivation mainly came from the recent theory of Generalized Hydrodynamics [1, 2], which aims to describe the non-equilibrium dynamics of integrable models on the Euler scale. One of the central elements of the theory is the set of continuity relations for the conserved charges of the models, because they lead to the generalized Euler-equations describing the ballistic flow of the quasi-particles. For this purpose it is essential to know the exact mean values of the currents associated to the conserved charges, assuming local equilibration on some intermediate length and time scales.

Regarding the thermodynamic limit an exact formula was postulated in [1,2], which was proven for relativistic QFT's in [1] (see also [3]). Regarding the spin current in the XXZ model the same formula was proven in [4]. The currents were also investigated in the classical Toda chain in [5]. Finally, in [6] the author and two collaborators derived an exact finite volume result for the current mean values, valid in a large class of Bethe Ansatz solvable quantum models. This derivation only uses a specific form factor expansion for the finite volume mean values, and the continuity equations that define the current operators. Therefore the proof applies to a number of models where that specific form factor expansion is established, including for example the Heisenberg spin chains, the Lieb-Liniger model, and the integrable QFT's with diagonal scattering. On the other hand, the extension to multi-component models solvable by the nested Bethe Ansatz is far from evident, due to the more complicated structure of the wave functions.

In this note we point out an interesting connection between the results of [6] and certain long-range deformations of the spin chains, that were discovered in the context of the AdS/CFT correspondence [7–9]. The common property of these long range spin chains is that they have a deformation parameter $\lambda$ (originating in the 't Hooft coupling of the CFT) and a commuting set of conserved charges, obtained as a power series in $\lambda$. The perturbations are local operators for each order in $\lambda$, albeit with growing range, and integrability (mutual commutativity) can be observed at each order in $\lambda$. And even though there are long range models with similar structures that can be defined in arbitrary finite volume (for example the Haldane-Sashtry model [10,11] and the Inozemtsev chain [12]), the deformations studied in [7–9] are strictly speaking only defined in infinite volume. For an introduction into these long range spin chains and their connection with the AdS/CFT we also recommend the review [13] and the paper [14].

The connection between the deformed chains and the current mean values is quite simple, and surprisingly it has not been noticed before: For each current operator there is a long-range deformation such that the given current operator itself is the leading perturbing operator. This provides a direct link towards the current mean values, which we explore in the present paper.

The paper is organized as follows. In Section 2 we fix our notations and discuss the concept of local and quasi-local operators. The local Heisenberg spin chain, its charge and current operators, and the main result of the previous work [6] are introduced in Section 3. In Section 4 we introduce the long-range deformations of the infinite XXX chain. The finite volume spectra of these deformed chains is studied in Section 5 which also includes our new derivation of the current mean values. Section 6 treats the $SU(3)$-symmetric fundamental model, for which we also derive the current mean values. Finally, we discuss our results and possible future directions in Section 7.

## 2   Local and quasi-local operators

In this work we will deal with spin chains with finite and infinite length. In the finite case the Hilbert space is

$$\mathcal{H} = \otimes_{j=1}^{L} \mathbb{C}^D. \tag{1}$$

In the concrete examples we will have $D = 2$ and $D = 3$. Furthermore we will always assume periodic boundary conditions in every finite volume situation.

We call an operator $\mathcal{O}(x)$ *local* if it acts only on a finite number of sites. Here it is understood that $\mathcal{O}(x)$ is the translation of $\mathcal{O}(0)$. We will denote by $|\mathcal{O}(x)|$ the range of the operator, i.e. the maximal number of neighboring sites on which it acts.

Extensive local operators can then be constructed as

$$\mathcal{O} = \sum_{x} \mathcal{O}(x), \tag{2}$$

where the summation runs over all sites of the chain.

The concept of *quasi-local operators* is very important. Following [15, 16] we call an extensive operator

$$A = \sum_{x} a(x) \tag{3}$$

quasi-local, if it satisfies the following two requirements:

1. The Hilbert-Schmidt norm of the operator grows at most linearly with the volume:

$$||A||^2 \sim L, \tag{4}$$

   where

$$||A||^2 \equiv \frac{\mathrm{Tr}\left(A^\dagger A\right)}{\mathrm{Tr}(1)}. \tag{5}$$

2. For every local operator $\mathcal{O}(x)$ the overlap with $A$ defined as

$$\langle A|\mathcal{O}(x)\rangle \equiv \frac{\mathrm{Tr}\left(A^\dagger \mathcal{O}(x)\right)}{\mathrm{Tr}(1)} \tag{6}$$

   has a finite $L \to \infty$ limit.

These conditions allow for arbitrary long range contributions to the densities $a(x)$, but their amplitudes have to decrease in some well defined way. Typically the amplitudes decay exponentially with the range.

The quasi-local operators play an essential role in the description of the Generalized Gibbs Ensemble of Heisenberg spin chain and related models [17–22]. There a commuting set of quasi-local charges is derived from the fused transfer matrices. These quasi-local charges form a complete set, which means that in the thermodynamic limit they determine the Bethe root densities of the equilibrium ensembles.

Below we will see that the quasi-local operators are central also to the present work: the long-range deformations naturally lead to quasi-local (but not local) charges.

## 3   Local charges and currents in the XXX chain

In this Section we concentrate on the Heisenberg spin chain given by the Hamiltonian

$$H^0_{XXX} = \sum_{j=1}^{L} (\sigma^x_j \sigma^x_{j+1} + \sigma^y_j \sigma^y_{j+1} + \sigma^z_j \sigma^z_{j+1} - 1). \tag{7}$$

In later Sections we will deal with deformed chains, therefore here and in the following the superscript $^0$ denotes that the corresponding operators and eigenvalues refer to the original, short range Hamiltonian.

It is known that the model possesses a set of local conserved charges in involution with each other:

$$[Q^0_\alpha, Q^0_\beta] = 0, \tag{8}$$

such that each charge has a local operator density

$$Q^0_\alpha = \sum_{x=1}^{L} q^0_\alpha(x). \tag{9}$$

The index $\alpha$ can be chosen such that $q^0_\alpha(x)$ spans $\alpha$ sites. For $\alpha = 1$ it is customary to use the global spin-$z$ operator. Furthermore, $Q_2 \sim H$ with a proportionality factor that depends on the conventions used. Here we define

$$q^0_2(x) = P_{x,x+1} - 1, \tag{10}$$

where $P$ is the permutation operator acting on $\mathbb{C}^2 \otimes \mathbb{C}^2$. This form has the advantage that it can be used also in the generic $SU(N)$-symmetric case. With this choice we have $H^0_{XXX} = 2Q^0_2$.

There are two standard methods to obtain the local conserved charges. One possibility is to derive them from the transfer matrix of the model, see for example [6]. The other possibility is with the use of the boost operator [23–26]. We now review this construction.

For any extensive local operator $L = \sum_{x=-\infty}^{\infty} l(x)$ let us define the boosted operator as the formal sum

$$\mathcal{B}[L] = \sum_{x=-\infty}^{\infty} x l(x). \tag{11}$$

The boosted operators are not homogeneous, and do not have a finite norm. Nevertheless they are very useful, because they can be used to generate new local and extensive quantities through formal commutation relations. For example, it is known that the higher conserved charges can be obtained as

$$Q^0_{\alpha+1} = i[\mathcal{B}[Q^0_2], Q^0_\alpha] + \text{constant}. \tag{12}$$

The constant part depends on the conventions used, and one way to fix it is by requiring that the eigenvalues on the reference state are all zero. For the equivalence of (12) with the transfer matrix construction we refer to [23–26].

For the range of the charges we have the following simple result:

$$|q^0_\alpha(x)| = \alpha, \tag{13}$$

which follows from the recursion above and $|q^0_2(x)| = 2$.

Also, it follows from the recursion that the charge densities can always be written as sums of exchange operators, that permute some subsets of the sites of the chain. Some explicit

formulas can be found in [9]. An alternative description using the spin generators was given in [27], including explicit representations for each charge density.

The main objects of this paper are the current operators, describing the flow of the conserved charges under unitary time evolution. For the time dependence of the charge contained in a finite interval the following continuity relation holds:

$$\frac{d}{dt} \sum_{x=x_1}^{x_2} q_\alpha^0(x) = i\left[H, \sum_{x=x_1}^{x_2} q_\alpha^0(x)\right] = J_\alpha^0(x_1) - J_\alpha^0(x_2+1). \tag{14}$$

Here the $J_\alpha^0(x)$ are the current operators associated to the charges $Q_\alpha^0$. Locality of the charges and the global commutation relations imply that (14) always has a solution with a local $J_\alpha^0(x)$ with range $|J_\alpha^0(x)| = \alpha + 1$. The defining relation in the most local form reads

$$i\left[H^0, q_\alpha^0(x)\right] = J_\alpha^0(x) - J_\alpha^0(x+1). \tag{15}$$

Let us now consider the charge contained in a half-infinite part of the chain. Then we get the formal definition

$$J_\alpha^0(x) = i\left[H^0, \sum_{y=x}^{\infty} q_\alpha^0(y)\right]. \tag{16}$$

The physical meaning is that the operator $J_\alpha^0(x)$ measures the current flowing into the right half-infinite part of the chain.

A formal summation of the above equation results in

$$\sum_{x=-\infty}^{\infty} J_\alpha^0(x) = i\left[H^0, \sum_{y=-\infty}^{\infty} x q_\alpha^0(x)\right] = i\left[H^0, \mathcal{B}[Q_\alpha^0]\right]. \tag{17}$$

The connection between (16) and (17) is only formal, because the complete summation of $J_\alpha^0(x)$ would result in infinite coefficients for the charge density. Nevertheless the subtraction of the global $Q_\alpha^0$ operator with formally infinite coefficient just removes this divergence, such that the remaining finite pieces have well defined values. This divergence problem is related to the question of where to set the zero coordinate for the boosted charge. Choosing an other point results in a finite difference of the global charge, which commutes with the Hamiltonian, thus not affecting the current operators. Alternatively, (17) can also be obtained by summing over (15) after multiplication by $x$.

It is also useful to define generalized current operators describing the flow of $Q_\alpha^0$ under time evolution dictated by $Q_\beta^0$. To this order we define

$$i\left[Q_\beta^0, q_\alpha^0(x)\right] = J_{\alpha,\beta}^0(x) - J_{\alpha,\beta}^0(x+1). \tag{18}$$

Locality of the charge densities and the global commutativity implies that the operator equation (18) can always be solved with some short-range $J_{\alpha,\beta}^0(x)$[1]. For the range we obtain the simple relation

$$|J_{\alpha,\beta}^0(x)| = |q_\alpha^0(x)| + |q_\beta^0(x)| - 1 = \alpha + \beta - 1. \tag{19}$$

In analogy with (17) we find the following relation for the generalized currents:

$$\sum_{x=-\infty}^{\infty} J_{\alpha,\beta}^0(x) = i[Q_\beta^0, \mathcal{B}[Q_\alpha^0]]. \tag{20}$$

---

[1]Our notation for the generalized currents slightly deviates from the one used in [6]: the operators $J_{\alpha,\beta}^0(x)$ defined here were denoted as $J_\alpha^\beta(x)$ there.

It is important that the charge densities and the corresponding currents are not well defined a priori. For each charge there is a "gauge freedom" for its density

$$q_\alpha^0(x) \quad \rightarrow \quad q_\alpha^0(x) + D(x+1) - D(x). \tag{21}$$

Such a transformation does not alter the integrated charges, but it changes the definition of the current operators as

$$J_{\beta,\alpha}^0(x) \quad \rightarrow \quad J_{\beta,\alpha}^0(x) - i[Q_\beta^0, D(x)]. \tag{22}$$

The mean values of the current operators in the eigenstates are not affected by this transformation, but the off-diagonal matrix elements do change. This has some consequences for the intermediate computations for the diffusive corrections in Generalized Hydrodynamics, which is discussed in detail in [28].

The general form of the charge densities and the relation (20) imply that the current operators also enjoy the $SU(2)$-invariance and they can also be expressed using exchange operators. However, as far as we know explicit results are not available for all $J_{\alpha,\beta}^0(x)$.

All of the above definitions involving the boost operators are defined in the infinite volume case. Nevertheless the charge densities and the generalized currents are perfectly well defined operators even in finite volume, as long as their range does not exceed the length of the chain.

It is our main goal to determine the exact finite volume mean values

$$\langle \psi | J_{\alpha,\beta}^0(x) | \psi \rangle \tag{23}$$

in all excited states of the finite volume systems. These mean values were computed in [6] using a form factor expansion. In the following we review these results.

## 3.1 Bethe Ansatz and the currents

The finite volume eigenstates of the Heisenberg spin chain can be found by the Bethe Ansatz [29]. The states are characterized by an ordered set of rapidities $\boldsymbol{\lambda}_N = \{\lambda_1, \dots, \lambda_N\}$ that describe the lattice momenta of the interacting spin waves. The un-normalized $N$-particle wave function can be written as

$$|\boldsymbol{\lambda}_N\rangle = \sum_{x_1 < x_2 < \dots < x_N} \sum_{\sigma \in S_N} \prod_{j>k} f(\lambda_{\sigma_j} - \lambda_{\sigma_k}) \prod_{j=1}^N e^{ip_{\sigma_j} x_j} |x_1, \dots, x_N\rangle, \tag{24}$$

where $|x_1, \dots, x_N\rangle$ are basis states with $N$ down spins at positions $x_j$, and it is understood that $p_j = p^0(\lambda_j)$ and $p^0(\lambda)$ is the one-particle propagation factor given explicitly by

$$e^{ip^0(\lambda)} = \frac{\lambda - i/2}{\lambda + i/2}. \tag{25}$$

The summation $\sigma \in S_N$ runs over all permutations of the rapidities. The function $f(\lambda) = 1 + i/\lambda$ is responsible for the interaction between the spin waves, such that the scattering factor becomes

$$S(\lambda) = e^{i\delta(\lambda)} = \frac{f(\lambda)}{f(-\lambda)} = \frac{\lambda + i}{\lambda - i}. \tag{26}$$

In this normalization the Bethe states are symmetric with respect to an exchange of rapidities.

These states are eigenvectors of the set of commuting charges. For the quasi-momentum we have

$$P^0 = \sum_{j=1}^N p^0(\lambda_j). \tag{27}$$

The energy eigenvalues are

$$E^0 = \sum_{j=1}^{N} e^0(\lambda_j), \qquad e^0(\lambda) = -\frac{2}{\lambda^2 + \frac{1}{4}}, \tag{28}$$

and for all higher charges

$$Q_\alpha^0 |\boldsymbol{\lambda}_N\rangle = \left[ \sum_{j=1}^{N} h_\alpha^0(\lambda_j) \right] |\boldsymbol{\lambda}_N\rangle, \tag{29}$$

where $h_\alpha^0(\lambda)$ are the one-particle eigenvalues. They satisfy the recursion

$$h_\alpha^0(\lambda) = -\partial_\lambda h_{\alpha-1}^0(\lambda) \tag{30}$$

and are given explicitly by

$$h_\alpha^0(\lambda) = i(\alpha - 2)! \left[ \frac{1}{(\lambda + i/2)^{\alpha-1}} - \frac{1}{(\lambda - i/2)^{\alpha-1}} \right]. \tag{31}$$

In finite volume the rapidities are subject to the Bethe equations, which guarantee periodicity of the wave function (24):

$$e^{ip(\lambda_j)L} \prod_{k \neq j} S(\lambda_j - \lambda_k) = 1, \qquad j = 1 \dots N. \tag{32}$$

For the Bethe states with on-shell rapidities the norm is [30]

$$\langle \boldsymbol{\lambda}_N | \boldsymbol{\lambda}_N \rangle = \left[ \prod_{j<k} f(\lambda_{jk}) f(\lambda_{kj}) \right] \times \det G. \tag{33}$$

Here $\det G$ is the Gaudin determinant, defined as follows. Let us write the Bethe equations in the logarithmic form:

$$p^0(\lambda_j)L + \sum_{k \neq j} \delta(\lambda_j - \lambda_k) = 2\pi I_j, \quad j = 1 \dots N. \tag{34}$$

Here $I_j \in \mathbb{Z}$ are the momentum quantum numbers, which can be used to parametrize the states. The Gaudin matrix is then defined as the Jacobian

$$G_{jk} = \frac{\partial}{\partial \lambda_k}(2\pi I_j), \qquad j, k = 1 \dots N, \tag{35}$$

where now the $I_j$ are regarded as functions of the rapidities. The explicit form is

$$G_{jk} = \delta_{jk} \left[ p'(\lambda_j)L + \sum_{l=1}^{N} \varphi(\lambda_j - \lambda_l) \right] - \varphi(\lambda_j - \lambda_k), \tag{36}$$

where

$$\varphi(\lambda) = \frac{\partial \delta(\lambda)}{\partial \lambda}. \tag{37}$$

For the normalized current mean values the following exact result was derived in [6]:

$$\frac{\langle \boldsymbol{\lambda}_N | J_\alpha^0(x) | \boldsymbol{\lambda}_N \rangle}{\langle \boldsymbol{\lambda}_N | \boldsymbol{\lambda}_N \rangle} = \mathbf{e}' \cdot G^{-1} \cdot \mathbf{h}_\alpha. \tag{38}$$

The quantities $\mathbf{e}'$ and $\mathbf{h}_\alpha$ are $N$-dimensional vectors given by

$$(\mathbf{e}')_j = \frac{\partial e^0(\lambda_j)}{\partial \lambda_j}, \qquad (\mathbf{h}_\alpha)_j = h_\alpha^0(\lambda_j), \tag{39}$$

and $G^{-1}$ is the inverse of the Gaudin matrix.

Similarly, for the generalized current operators the following was obtained in [6]:

$$\frac{\langle \boldsymbol{\lambda}_N | J_{\alpha,\beta}^0(x) | \boldsymbol{\lambda}_N \rangle}{\langle \boldsymbol{\lambda}_N | \boldsymbol{\lambda}_N \rangle} = \mathbf{h}_\beta' \cdot G^{-1} \cdot \mathbf{h}_\alpha. \tag{40}$$

As special cases we have

$$\frac{\langle \boldsymbol{\lambda}_N | J_{2,\beta}^0(x) | \boldsymbol{\lambda}_N \rangle}{\langle \boldsymbol{\lambda}_N | \boldsymbol{\lambda}_N \rangle} = -\frac{1}{L} \frac{\langle \boldsymbol{\lambda}_N | Q_{\beta+1}^0 | \boldsymbol{\lambda}_N \rangle}{\langle \boldsymbol{\lambda}_N | \boldsymbol{\lambda}_N \rangle}, \tag{41}$$

which is in accordance with the boost relation (12).

The current mean values display the symmetry

$$\frac{\langle \boldsymbol{\lambda}_N | J_{\alpha,\beta}^0(x) | \boldsymbol{\lambda}_N \rangle}{\langle \boldsymbol{\lambda}_N | \boldsymbol{\lambda}_N \rangle} = \frac{\langle \boldsymbol{\lambda}_N | J_{\beta+1,\alpha-1}^0(x) | \boldsymbol{\lambda}_N \rangle}{\langle \boldsymbol{\lambda}_N | \boldsymbol{\lambda}_N \rangle}, \tag{42}$$

which follows from the symmetry of the Gaudin-matrix and the recursion (30).

There are two possible interpretations of the result (40), both using the derivative of the Bethe equations with respect to certain parameters.

The first interpretation was already given in [6]: The mean values can be written as

$$\frac{\langle \boldsymbol{\lambda}_N | J_{\alpha,\beta}^0(x) | \boldsymbol{\lambda}_N \rangle}{\langle \boldsymbol{\lambda}_N | \boldsymbol{\lambda}_N \rangle} = \frac{1}{L} \sum_{j=1}^{N} q_\alpha^0(\lambda_j) v_{\text{eff}}^\beta(\lambda_j), \tag{43}$$

where the quantities

$$v_{\text{eff}}^\beta(\lambda_j) = L(G^{-1}\mathbf{h}_\beta')_j = \frac{L}{2\pi} \frac{\partial Q_\beta^0}{\partial I_j} \tag{44}$$

are interpreted as effective velocities, describing particle propagation under time evolution by $Q_\beta^0$. In [6] the effective velocities were only treated in the case when time evolution is generated by the physical Hamiltonian, but the extension to the flow generated by $Q_\beta^0$ is straightforward.

There is a simple semi-classical picture explaining the above formula for $v_{\text{eff}}$, laid out in [6]. Here we summarize the main points. In a free model the Gaudin matrix is diagonal and we would get the standard expression for the group velocities of wave packets:

$$v_{\text{eff}}^\beta(\lambda) = \frac{\partial h_\beta^0(\lambda)/\partial \lambda}{\partial p(\lambda)/\partial \lambda} = \frac{dh_\beta^0}{dp}. \tag{45}$$

In the presence of interactions this simple one-particle result is modified, such that the effective velocity also takes into account the displacement of the wave packets suffered during the scattering processes. The inverse of the Gaudin matrix then emerges from a self-consistent computation for the average velocities [6].

The second interpretation of (40) is new and it serves as the motivation for the present paper. It is explained below.

### 3.2 Strategy towards the deformed spin chains

Let us consider an artificial set of Bethe equations with a deformation parameter $\kappa$:

$$(p^0(\lambda_j) + \kappa h^0_\alpha(\lambda_j))L + \sum_{k \neq j} \delta(\lambda_j - \lambda_k) = 2\pi I_j \quad j = 1 \ldots N. \tag{46}$$

Here the $\kappa$-dependent term can be interpreted as a momentum-dependent twist. We will be looking for a spatially homogeneous deformation of the spin chain that leads to the above Bethe equations.

If we fix the momentum quantum numbers for a given solution, then the Bethe roots will become functions of $\kappa$. Let us compute the change of the mean value of a given charge, caused by the change in the rapidities. Using the formula (29) we get the relation

$$\frac{dQ^0_\beta}{d\kappa} = \sum_{j=1}^{N} \frac{\partial h^0_\beta(\lambda_j)}{\partial \lambda_j} \frac{d\lambda_j}{d\kappa} = L\left(\mathbf{h}'_\beta \cdot G^{-1} \cdot \mathbf{h}_\alpha\right). \tag{47}$$

The coincidence between (47) and (40) suggests that we should look for a perturbation of the charge $Q^0_\beta$ as

$$Q^\kappa_\beta = Q^0_\beta + \kappa \sum_{x=1}^{L} J^0_{\alpha,\beta}(x), \tag{48}$$

and investigate the $\kappa$-dependence of its mean value. The result (40) could then follow from the Hellmann-Feynman theorem, *if*

- integrability holds after the perturbation

- the only effect of the deformation is the appearance of the $\kappa$-dependent terms in the Bethe equations (46)

- the charge eigenvalues can be computed using the same formula (29), but with the deformed Bethe roots.

Luckily, there are deformations that *almost* guarantee the above requirements, and they have already been explored in the context of the AdS/CFT correspondence [9]. Here we describe the main ideas behind the construction, and the details will be presented in the next section.

In infinite volume it is possible to deform the set of conserved charges in many ways so that each charge becomes a power series in $\kappa$ and the set remains commutative:

$$Q^\kappa_\alpha = Q^0_\alpha + \sum_{j=1}^{\infty} \frac{\kappa^j}{j!} Q^{(j)}_\alpha. \tag{49}$$

Furthermore, it is possible to choose the deformations such that at linear order in $\kappa$ we have indeed (48). In this process the commutation relations

$$[Q^\kappa_\alpha, Q^\kappa_\beta] = 0 \tag{50}$$

can be ensured at every order in $\kappa$ recursively. The range of the deforming operators grows linearly with the order in $\kappa$. It follows that the summation of the perturbation series will not be local: instead we can assume to get quasi-local operators.

However, there is an important complication: the deformation is only defined in infinite volume. As we will see below, these deformations are not generated from a local transfer matrix construction, therefore they can not be uniquely defined in finite volume. In fact, in

finite volume the perturbation series for the charges is valid only as long as the correction terms fit into the given volume. If the range of the perturbing operators exceeds the length of the chain at a given order, then we are faced with the *wrapping problem*: it is generally not known how to put the higher order terms into the finite volume. Thus we can not ensure integrability at the higher orders in $\kappa$. In this case the perturbation series (49) has to be truncated at some order, and the truncation level will depend on $L$ and the index $\alpha$.

The key idea is that even though integrability is ensured only up to the lower orders in $\kappa$, even the finite volume system can be solved with integrability techniques, up to the given order in $\kappa$. The goal is to express all physical quantities (such as energy and charge eigenvalues) in a perturbation series in $\kappa$, and build the Bethe Ansatz using the infinite volume quantities, including corrections up to given order in $\kappa$, or possibly using an all orders result. This method is called the *asymptotic Bethe Ansatz*, because it becomes asymptotically exact if we fix the particle number and then take the $L \to \infty$ limit. In a finite volume the asymptotic results should be trusted up to a certain order in $\kappa$, for which the commutativity of some subset of the charges still holds in that given volume.

In Section 5 we will argue that this asymptotic procedure does in fact yield the exact mean values of the current operators.

# 4   Long range deformations

In this Section we consider the infinite volume situation. Following [9] we introduce certain long range deformations of the original Heisenberg model, such that integrability remains preserved. Denoting the deformation parameter by $\kappa$ it is our goal to find a set of operators $\{Q_\alpha^\kappa\}_{\alpha=1,2,...}$ such that

$$[Q_\alpha^\kappa, Q_\beta^\kappa] = 0, \tag{51}$$

and

$$Q_\alpha^{\kappa=0} = Q_\alpha^0. \tag{52}$$

We require that the resulting operators should be extensive and quasi-local:

$$Q_\alpha^\kappa = \sum_{x=-\infty}^{\infty} q_\alpha^\kappa(x). \tag{53}$$

Quasi-locality implies that the operator density $q_\alpha^\kappa(x)$ has to have a finite norm.

One way to obtain such long range deformations is by postulating a generating equation for the charges, that describes "evolution in $\kappa$". This takes the Lax form

$$\frac{d}{d\kappa} Q_\alpha^\kappa = i[X(\kappa), Q_\alpha^\kappa]. \tag{54}$$

Here $X(\kappa)$ is a formal operator that will be specified below. It is important that in general $X$ depends on $\kappa$.

This deformation leaves the commutation relations between the charges unmodified, which follows from the Jacobi identity

$$\frac{d}{d\kappa}[Q_\alpha^\kappa, Q_\beta^\kappa] = i[X(\kappa), [Q_\alpha^\kappa, Q_\beta^\kappa]]. \tag{55}$$

Commuting charges will be mapped to commuting charges, furthermore the group theoretical properties (commutation relations between some Lie algebra generators) will also be preserved.

We note that our $\kappa$ parameter is not identical to the $\lambda$ deformation parameter used in [9]. We choose our deformations such that the desired current operators always appear in linear order in $\kappa$. In contrast, in the formalism of [9] they would only appear in higher orders in $\lambda$.

The generating equation suggests a transformation rule also for the eigenstates of the model. Let $\left|\Psi^0\right\rangle$ be any eigenstate of the set of un-deformed charges of the infinite volume model. Then we define the deformation of the state as

$$\frac{d}{d\kappa}|\Psi^\kappa\rangle = -iX(\kappa)|\Psi^\kappa\rangle, \tag{56}$$

with the initial condition $|\Psi(\kappa=0)\rangle = \left|\Psi^0\right\rangle$. Then the deformation relations imply that eigenstates are mapped to eigenstates and the charge eigenvalues $\Lambda_\alpha^0$ are not deformed:

$$Q_\alpha^\kappa|\Psi^\kappa\rangle = \Lambda_\alpha^0|\Psi^\kappa\rangle. \tag{57}$$

We stress however that these relations are only formal, and the question whether they are well defined and physically acceptable depends strongly on the structure of $X$. The main goal is to find some formal expressions for $X$ that generate quasi-local deformations.

In [9] three families for $X$ were identified (with an additional possibility for the deformation, see below):

### 1: Local and quasi-local operators.

The choice

$$X = \sum_{x=\infty}^{\infty} \mathcal{O}(x), \tag{58}$$

with $\mathcal{O}$ being a short range operator describes a "physical" similarity transformation. In this case $X$ does not depend on $\kappa$ and it could be regarded as the Hamiltonian of some other model, which simply generates a change of basis for our model:

$$Q_\alpha^\kappa = e^{i\kappa X} Q_\alpha^0 e^{-i\kappa X}. \tag{59}$$

The generating equation and the transformation can be defined also in finite volume $L$, as long as $|\mathcal{O}(x)| \leq L$. The deformation leaves the spectrum of all charges unmodified, in every finite volume.

The same construction can be extended to the case when $X$ is a quasi-local operator. In that case the transformation is strictly defined only in infinite volume, nevertheless it can be understood as a similarity transformation.

### 2: Boost operators.

The choice

$$X(\kappa) = -\mathcal{B}[Q_\alpha^\kappa] \tag{60}$$

for some $\alpha$ also generates quasi-local charges. Now the $\kappa$-derivative of the charges is given by

$$\frac{d}{d\kappa}Q_\beta^\kappa = J_{\alpha,\beta}^\kappa, \tag{61}$$

where we defined the $\kappa$-deformed generalized current operators as

$$J_{\alpha,\beta}^\kappa = -i[\mathcal{B}[Q_\alpha(\kappa)], Q_\beta(\kappa)]. \tag{62}$$

The global commutativity of the charges imply that $J^{\kappa}_{\alpha,\beta}$ is an extensive operator, this can be shown recursively for every order in $\kappa$. The full operator is expected to be quasi-local. The minus sign in (60) is chosen only for later convenience.

Let us define the deformed current densities as

$$i\left[Q^{\kappa}_{\beta}, q^{\kappa}_{\alpha}(x)\right] = J^{\kappa}_{\alpha,\beta}(x) - J^{\kappa}_{\alpha,\beta}(x+1). \tag{63}$$

Similar arguments as above show that

$$J^{\kappa}_{\alpha,\beta} = \sum_{x=-\infty}^{\infty} J^{\kappa}_{\alpha,\beta}(x). \tag{64}$$

It is important that the simple choice $X(\kappa) = -\mathcal{B}[Q^{0}_{\alpha}]$ would not guarantee quasi-locality for higher order corrections in $\kappa$, and it is important to involve the solution $Q^{\kappa}_{\alpha}$ in the generator itself.

The generating equation is not a well-defined differential equation for the charges, because the boost operators depend strongly on how the charge densities are chosen. A "gauge transformation" of the form (21) results in

$$\mathcal{B}[Q^{\kappa}_{\alpha}] \quad \rightarrow \quad \mathcal{B}[Q^{\kappa}_{\alpha}] + \sum_{x} D(x). \tag{65}$$

The two boost operators thus differ in a local operator, whose additional effect during the deformation has to be compensated by a similarity transformation. Nevertheless these additional pieces leave both the finite and infinite volume spectra of the charges invariant, therefore they are irrelevant for our purposes.

We note that the convergence of the deformation series and the quasi-locality of the resulting sum has not yet been proven rigorously. We expect that the deformed charges will be quasi-local in some finite neighborhood of $\kappa = 0$.

### 3: Bi-local operators.

Let $A$ and $B$ be two quasi-local operators given by the operator densities $A(x)$ and $B(x)$. We then define the bi-local operator

$$[A|B] \equiv \sum_{x<y} \{A(x), B(y)\} + \frac{1}{2}\sum_{x}\{A(x), B(x)\}. \tag{66}$$

Note that the definition immediately yields the relation

$$[A|B] + [B|A] = \{A, B\}. \tag{67}$$

The bi-local operator is thus "one-half" of the anti-commutator, corresponding to a specific spatial ordering of the operator densities.

The long-range deformations can be generated by the operators

$$X(\kappa) = [Q^{\kappa}_{\alpha}|Q^{\kappa}_{\beta}]. \tag{68}$$

Once more the global commutativity of the charges imply that the resulting deformation will be quasi-local [9]. Our definition (66) differs slightly from the one in [9], where the relative positions of the charge densities are shifted as compared to our formula. Nevertheless the difference between the two choices is just an extensive local operator. Similarly, the gauge transformations (21) can also lead to additional extensive local operators.

**4: Basis change for the charges.**

There is a further possibility for the deformation of the charges, namely a linear mixing of the set $\{Q_\alpha^\kappa\}$, with a mixing matrix that can depend on $\kappa$. Such transformations are essential for the spin chains relevant to AdS/CFT [7–9], but we will not use this possibility.

In the following we will focus on the case when the generator $X(\kappa)$ is chosen to be one of the boost operators.

## 4.1   Deformation of eigenstates

It is useful to study the deformation of the states, which is generated by the relation (56). Using (60) we have:

$$\frac{d}{d\kappa}|\Psi(\kappa)\rangle = i\mathcal{B}[Q_\alpha(\kappa)]|\Psi(\kappa)\rangle. \tag{69}$$

We first analyze the effect of the deformation on the global momentum of the states. Let $U = e^{iP}$ be the one-site translation operator, which satisfies

$$U\mathcal{O}(x) = \mathcal{O}(x+1)U \tag{70}$$

for any local operator.

Let us denote the eigenvalue of $U$ on some un-deformed eigenstate $\left|\Psi^0\right\rangle$ of the infinite volume system as $e^{iP_0}$. Our deformations produce extensive and homogeneous charges, thus we can assume that the deformed eigenstates $|\Psi(\kappa)\rangle$ will remain eigenstates of $U$ for all $\kappa$ (this also follows from the explicit form of the generating equation). We are thus looking for the deformed eigenvalues

$$U|\Psi(\kappa)\rangle = e^{iP(\kappa)}|\Psi(\kappa)\rangle. \tag{71}$$

Taking a formal derivative in $\kappa$ we get

$$U\frac{d}{d\kappa}|\Psi(\kappa)\rangle = iP'(\kappa)e^{iP(\kappa)}|\Psi(\kappa)\rangle + e^{iP(\kappa)}\frac{d}{d\kappa}|\Psi(\kappa)\rangle, \tag{72}$$

which can equivalently be written as

$$UX|\Psi(\kappa)\rangle + \frac{dP(\kappa)}{d\kappa}e^{iP(\kappa)}|\Psi(\kappa)\rangle = e^{iP(\kappa)}X|\Psi(\kappa)\rangle. \tag{73}$$

Let us now act with $X = -\mathcal{B}[Q_\alpha(\kappa)]$ on the two sides of (71), resulting in

$$-\mathcal{B}[Q_\alpha(\kappa)]U|\Psi(\kappa)\rangle = (-U\mathcal{B}[Q_\alpha(\kappa)] + UQ_\alpha(\kappa))|\Psi(\kappa)\rangle = -e^{iP(\kappa)}\mathcal{B}[Q_\alpha(\kappa)]|\Psi(\kappa)\rangle. \tag{74}$$

Comparing these equations we can read off

$$\frac{d}{d\kappa}P(\kappa) = \Lambda_\alpha. \tag{75}$$

As it was discussed above, the eigenvalues of the charges do not change under the deformation, thus we get the simple solution

$$P(\kappa) = P^0 + \Lambda_\alpha\kappa. \tag{76}$$

This is the generalization of the standard boost operation known in Lorentzian of Galilean invariant models. It holds for all eigenstates of the infinite volume system.

On one-particle states the deformation results in the modified momentum-rapidity relation

$$p^0(\lambda) \quad \rightarrow \quad p^0(\lambda) + \kappa h_\alpha^0(\lambda). \tag{77}$$

The quasi-locality of the deformation implies that the multi-particle momentum of Bethe states has to be of the form

$$P^\kappa = \sum_{j=1}^N p^\kappa(\lambda_j), \qquad p^\kappa(\lambda_j) = p^0(\lambda) + \kappa h_\alpha^0(\lambda_j). \tag{78}$$

The deformation equation (56) also implies that the for well separated particles the deformed eigenstates will also take the form of a Bethe state, with the modified dispersion:

$$|\Psi^\kappa(\lambda_N)\rangle \approx \sum_{x_1 < x_2 < \cdots < x_N} \sum_{\sigma \in S_N} e^{ip^\kappa(\lambda_{\sigma j})x_j} \prod_{j<k} f(\lambda_{\sigma_j} - \lambda_{\sigma_k}). \tag{79}$$

Here it is understood that the wave function obtains corrections in various orders in $\kappa$ depending on the distances $|x_j - x_k|$, but this form becomes exact in the limit $|x_j - x_k| \to \infty$. The additional pieces are called *contact terms* and arise from the long-range terms in (56).

The boost operator is a one-particle irreducible operator, and it does not change the relative phases dictated by $f(\lambda)$. These can be deformed if we choose $X(\kappa)$ to be a bi-local operator, because the bi-local generators include a specific spatial ordering for the generating charges, thus distinguishing the terms in the Bethe wave functions with different particle orderings [9]. We give a few comments about this case in Section 7.

## 5 Asymptotic Bethe Ansatz and the current mean values

Here we investigate the finite volume spectrum of the deformed charges, in the case of the boost deformations. Our motivation is that if the exact charge eigenvalues are known as a function of $\kappa$, then the Hellmann-Feynman theorem together with (61) implies

$$\left\langle \Psi^0 | J_{\alpha,\beta}^0(x) | \Psi^0 \right\rangle = \frac{1}{L} \left. \frac{dQ_\beta^\kappa}{d\kappa} \right|_{\kappa=0}. \tag{80}$$

As we explained above, this problem is strictly speaking not well defined: generally it is not known how to define the charges to all orders in $\kappa$ in a finite volume, due to the *wrapping problem*. Nevertheless, low order results in $\kappa$ can be obtained even in finite volume, and we are only interested in the linear terms. This justifies our procedure.

In the previous Section we explained that in infinite volume the eigenvalues of the charges are not modified during the deformation. However, this is not true anymore in finite volume. The main reason for this is that in finite volume the rapidities undergo a deformation, and this can be understood using the *asymptotic Bethe Ansatz*. The main idea is to use the approximate wave function (79) and to derive the Bethe equations using the deformed one-particle momenta. This procedure becomes exact in the $L \to \infty$ limit, with fixed $N$. Furthermore, we will argue that it leads to the correct results for the current mean values even when $L$ is comparable to $N$.

For a given state with momentum quantum numbers $\{I_j\}$ the asymptotic Bethe equations for our deformation read

$$p^\kappa(\lambda_j)L + \sum_{k \neq j} \delta(\lambda_j - \lambda_k) = 2\pi I_j, \tag{81}$$

with the deformed momentum given by (78).

The first order correction to the rapidities is easily computed by taking derivatives of the above equations, leading to

$$\frac{\partial \lambda_j}{\partial \kappa} = L G^{-1} \mathbf{h}_\alpha, \tag{82}$$

where $\mathbf{h}_\alpha$ is a vector with elements $h_\alpha(\lambda_j)$. Computing finally (80) gives

$$\left\langle \Psi^0 \middle| J^0_{\alpha,\beta}(x) \middle| \Psi^0 \right\rangle = \sum_{j=1}^{N} h'_\beta(\lambda_j) \frac{\partial \lambda_j}{\partial \kappa} = \mathbf{h}'_\beta G^{-1} \mathbf{h}_\alpha. \tag{83}$$

This result is valid as long as there are no wrapping corrections involved, which means that the range of current $J^0_{\beta,\alpha}$ must not exceed the length of the spin chain.

With this we have re-derived the main result of [6].

The alerted reader might have the following objection to our derivation: In those cases when the perturbing operator barely fits into the volume (for example $\alpha + \beta - 1 = L$), or when $N$ is comparable to $L$, our arguments for the asymptotic Bethe Ansatz break down, thus the above result is not justified. Nevertheless we can argue that it is still valid. The reason lies in the structure of the mean values, when viewed as a function of $L$ and the rapidities, for a fixed particle number.

When the mean value of any local operator is computed using the exact original Bethe Ansatz wave function, the result can always be expressed as a rational function of $L$ and $\boldsymbol{\lambda}_N$. Focusing on the $L$-dependence, the rational function can be expanded into a convergent power series in $1/L$. We claim that for the current mean values the coefficients of this series can be determined from large enough volumes, using our computation with the asymptotic Bethe Ansatz. This is possible because the error terms to the asymptotic results decay exponentially with the volume, as they are always proportional to $\kappa^c$ with $c \sim L$. Therefore our perturbative computation can indeed fix all polynomial terms in $1/L$, which completely fixes the rational function in question. The results are thus correct, as long as the real space computation of the mean value is well defined, i.e. as long as the current operators fit into the given volume.

## 5.1 Inhomogeneous spin chains

It is known that the boost deformations treated here can be approximated to a certain order in $\kappa$ by introducing an inhomogeneous spin chain, and adjusting the inhomogeneities in a special way [9]. This is possible because the one-particle propagation on the spin chain depends on the inhomogeneities, and the deformed dispersion relations can be approximated with a given precision, depending on the length of the chain. Furthermore, the inhomogeneities can also be used to generate the conserved operators and the eigenstates of the deformed spin chains relevant to AdS/CFT [31–34].

We do not investigate the inhomogeneous chains here, because for our purposes it is sufficient to know the deformations of the Bethe equations, from which the mean values immediately follow. We just remark that the inhomogeneities could be used to get explicit real space formulas for the current operators. This is left to further work.

# 6 $SU(3)$-symmetric model

The results of the previous sections admit simple generalizations to models with higher rank symmetries, solvable by the nested Bethe Ansatz. Here we will focus on the $SU(3)$-symmetric fundamental model given by the Hamiltonian

$$H^0 = \sum_{x=1}^{L} (P_{x,x+1} - 1), \tag{84}$$

where now $P$ is the permutation operator acting on $\mathbb{C}^3 \otimes \mathbb{C}^3$.

Just like in the previous case, higher charges of the model can be constructed using the boost operator. We now choose $Q_2^0 = H^0$, thus $q_2^0(x)$ is the same as in the $SU(2)$-case, see (10). With this choice *all* local charges of the model will take the same form as in the $SU(2)$ case, when expressed using the permutation operators. A number of explicit cases are given in [9].

We define the generalized current operators $J_{\alpha,\beta}^0$ with the same continuity equations as before, and they can be expressed using the boost operators as in (20). When expressed using permutation operators, they will take the exact same form as in the XXX model. Thus the only formal difference compared to the $SU(2)$-case lies in the construction of the eigenstates.

In the $SU(3)$-case let $|0\rangle$, $|1\rangle$ and $|2\rangle$ be a basis of $\mathbb{C}^3$ and let us choose the reference state as

$$|\emptyset\rangle = \otimes_{j=1}^L |0\rangle, \tag{85}$$

which is an eigenstate of the Hamiltonian. Excitations can then be created over this reference state, and the one-particle states have an inner degree of freedom corresponding to the available directions $|1\rangle$ and $|2\rangle$. Similarly, the $N$-particle states can be characterized by an orientation vector which is an element of $\otimes_{j=1}^N \mathbb{C}^2$. The exact finite volume eigenstates of the model can be constructed using the *nested Bethe Ansatz*. The resulting $N$-particle states are characterized by the set $\lambda_N$ describing the lattice momenta of the particles, and a further set of auxiliary rapidities $\mu_M$ describing the orientation in the internal space. We will use the notation $|\lambda_N, \mu_M\rangle$; for their construction see [35]. The $GL(3)$ quantum numbers of the state $|\lambda_N, \mu_M\rangle$ are given by $(L-N, N-M, M)$.

The full set of rapidities is subject to the nested Bethe equations

$$e^{ip^0(\lambda_j)L} \prod_{k=1,k\neq j}^N S(\lambda_j - \lambda_k) \prod_{k=1}^M \tilde{S}(\lambda_j - \mu_k) = 1 \qquad j = 1, \ldots, N$$

$$\prod_{k=1}^M \tilde{S}(\mu_j - \lambda_k) \prod_{k=1,k\neq j}^N S(\mu_j - \mu_k) = 1, \qquad j = 1, \ldots, M, \tag{86}$$

where $p^0(\lambda)$ and $S(\lambda)$ are given by (25) and (26), respectively, and

$$\tilde{S}(\lambda) = e^{i\tilde{\delta}(\lambda)} = \frac{\lambda - \frac{i}{2}}{\lambda + \frac{i}{2}}. \tag{87}$$

The lattice momentum, the energy, and the higher charge eigenvalues are given by the sum of one particle eigenvalues of the first type of rapidities:

$$P^0 = \sum_{j=1}^N p^0(\lambda_j), \qquad E = \sum_{j=1}^N h_2^0(\lambda_j), \qquad Q_\alpha^0 = \sum_{j=1}^N h_\alpha^0(\lambda_j). \tag{88}$$

Let us write the nested Bethe equations in the logarithmic form as

$$p^0(\lambda_j)L + \sum_{k=1,k\neq j}^N \delta(\lambda_j - \lambda_k) + \sum_{k=1}^M \tilde{\delta}(\lambda_j - \mu_k) = 2\pi I_j, \quad j = 1 \ldots N,$$

$$\sum_{k=1}^N \tilde{\delta}(\mu_j - \lambda_k) + \sum_{k=1,k\neq j}^M \delta(\mu_j - \mu_k) = 2\pi I_{N+j}, \qquad j = 1 \ldots M. \tag{89}$$

Analogously to the $SU(2)$-case we define the Gaudin matrix of size $(N+M) \times (N+M)$ as the Jacobian of the mapping from the rapidities to the integer quantum numbers:

$$G_{jk} = \frac{\partial(2\pi I_j)}{\partial u_k}, \tag{90}$$

where it is understood that

$$\{\mathbf{u}_{N+M}\} = \{\lambda_1, \ldots, \lambda_N, \mu_1, \ldots, \mu_M\}. \tag{91}$$

## 6.1  Boost deformation and current mean values

Similar to the previous case we perform the deformation procedure with the boost operators given by (60).

It can be argued [9] that even in this case the only change in the infinite volume states (for large enough separation of the particles) is the replacement

$$p^0(\lambda) \quad \to \quad p^0(\lambda) + \kappa h_\alpha^0(\lambda). \tag{92}$$

The spin structure of the infinite volume states is not affected by the boost transformation.

In finite volume the deformation of the nested Bethe equations derived from the *asymptotic Bethe Ansatz* are

$$
(p^0(\lambda_j) + \kappa h_\alpha^0(\lambda_j))L + \sum_{k=1, k\neq j}^{N} \delta(\lambda_j - \lambda_k) + \sum_{k=1}^{M} \tilde{\delta}(\lambda_j - \mu_k) = 2\pi I_j, \quad j = 1\ldots N,
$$
$$
\sum_{k=1}^{N} \tilde{\delta}(\mu_j - \lambda_k) + \sum_{k=1, k\neq j}^{M} \delta(\mu_j - \mu_k) = 2\pi I_{N+j}, \qquad j = 1\ldots M. \tag{93}
$$

Note that the only change is the modified physical dispersion relation for the physical rapidities, and the equations for the auxiliary $\boldsymbol{\mu}_M$ are not changed. Nevertheless, the actual solutions for $\boldsymbol{\mu}_M$ in a given volume $L$ are deformed, due to the coupling with $\boldsymbol{\lambda}_N$.

We now apply the Hellmann-Feynman theorem to find the mean values of the generalized charges. A straightforward computation, completely analogous to the $SU(2)$-case now yields

$$\frac{\langle \boldsymbol{\lambda}_N, \boldsymbol{\mu}_M | J_{\alpha,\beta}^0(x) | \boldsymbol{\lambda}_N, \boldsymbol{\mu}_M \rangle}{\langle \boldsymbol{\lambda}_N, \boldsymbol{\mu}_M | \boldsymbol{\lambda}_N, \boldsymbol{\mu}_M \rangle} = \tilde{\mathbf{h}}_\beta' \cdot G^{-1} \cdot \tilde{\mathbf{h}}_\alpha, \tag{94}$$

where $G^{-1}$ is the inverse of the Gaudin matrix of size $(N + M) \times (N + M)$, and $\tilde{\mathbf{h}}_\alpha$ is a vector of length $N + M$ with components

$$(\tilde{\mathbf{h}}_\alpha)_j = \begin{cases} h_\alpha^0(\lambda_j) & j = 1\ldots N \\ 0 & j = (N+1)\ldots(N+M), \end{cases} \tag{95}$$

and $\tilde{\mathbf{h}}_\beta'$ is defined analogously.

This result can also be formulated using a "direct" Gaudin matrix of size $N \times N$. Let us regard the Bethe equations as a mapping between the variables $\{2\pi I_j\}_{j=1,\ldots N+M}$ and $\{\mathbf{u}_{N+M}\} = \{\boldsymbol{\lambda}_N, \boldsymbol{\mu}_M\}$. We assume that this mapping is bijective in some neighborhood of the Bethe state, and for the inverse of the Gaudin matrix we have

$$(G^{-1})_{jk} = \frac{\partial u_j}{\partial(2\pi I_k)}. \tag{96}$$

According to (94) only the upper left $N \times N$ block of this matrix is needed for the current mean values. This block describes the change of the first set of rapidities with respect to their momentum quantum numbers, while keeping the quantum numbers for the $\mu$-variables fixed. This means that as we change the $I_j$, $j = 1, \ldots, N$, it is implicitly assumed that the $\mu$ variables

are also changed, and this is substituted back into the Bethe equations of the first set. Thus we obtain

$$\frac{\langle \boldsymbol{\lambda}_N, \boldsymbol{\mu}_M | J_{\alpha,\beta}^0(x) | \boldsymbol{\lambda}_N, \boldsymbol{\mu}_M \rangle}{\langle \boldsymbol{\lambda}_N, \boldsymbol{\mu}_M | \boldsymbol{\lambda}_N, \boldsymbol{\mu}_M \rangle} = \mathbf{h}_\beta' \cdot G_{\text{direct}}^{-1} \cdot \mathbf{h}_\alpha, \tag{97}$$

where now $\mathbf{h}_\alpha$ and $\mathbf{h}_\beta'$ are vectors of length $N$, given by the one-particle charge eigenvalues, and $G_{\text{direct}}^{-1}$ is a matrix of size $N \times N$ defined as

$$\left( G_{\text{direct}}^{-1} \right)_{jk} = \left. \frac{\partial \lambda_j}{\partial (2\pi I_k)} \right|_{I_{N+1},\dots,I_{N+M} \text{ fixed}}. \tag{98}$$

A somewhat more explicit formula can be given if we write the original Gaudin matrix in the block form

$$G = \begin{pmatrix} A & B \\ B^T & C \end{pmatrix}, \tag{99}$$

where $A$, $B$ and $C$ are matrices of size $N \times N$, $N \times M$ and $M \times M$, and $B^T$ is the transpose of $B$. Then the direct Gaudin matrix is

$$G_{\text{direct}} = A - B C^{-1} B^T. \tag{100}$$

Interestingly, such "direct" Jacobians appeared in early work dealing with the norm of the Bethe states, see for example eq. (24) in [36].

Equations (94) and (97) are the main result for the current operators in the $SU(3)$-symmetric chain. Similar to the $SU(2)$-case they allow a semi-classical interpretation. We can write the mean values as

$$\frac{\langle \boldsymbol{\lambda}_N, \boldsymbol{\mu}_M | J_{\alpha,\beta}^0(x) | \boldsymbol{\lambda}_N, \boldsymbol{\mu}_M \rangle}{\langle \boldsymbol{\lambda}_N, \boldsymbol{\mu}_M | \boldsymbol{\lambda}_N, \boldsymbol{\mu}_M \rangle} = \frac{1}{L} \sum_{j=1}^{N} v_{\text{eff}}^\beta(\lambda_j) h_\alpha^0(\lambda_j), \tag{101}$$

where now the effective velocities are given by the first $N$ components of the vector $L G^{-1} \tilde{\mathbf{h}}_\beta'$, or by the components of the vector $L G_{\text{direct}}^{-1} \mathbf{h}_\beta'$. Alternatively they can be expressed as

$$v_{\text{eff}}^\beta(\lambda_j) = L \left. \frac{\partial Q_\beta^0}{\partial (2\pi I_j)} \right|_{I_{N+1},\dots,I_{N+M} \text{ fixed}}. \tag{102}$$

In the semi-classical picture they describe the particle propagation in the presence of the other particles, for the given orientation of the internal degrees of freedom. Identifying $2\pi I_j / L$ as a "dressed momentum" $p_j^{\text{dr}}$ we also have the suggestive relation

$$v_{\text{eff}}^\beta(\lambda_j) = \left. \frac{\partial Q_\beta^0}{\partial p_j^{\text{dr}}} \right|_{I_{N+1},\dots,I_{N+M} \text{ fixed}}. \tag{103}$$

It is important that the effective velocities always depend on the second set of rapidities $\boldsymbol{\mu}_M$ as well, even though this is implicit in the above formulas.

# 7 Discussion

In this work we derived the mean values of the current and generalized current operators in the $SU(2)$ and $SU(3)$-symmetric fundamental models, with periodic boundary conditions.

The key observation was that these operators appear as the first order perturbation terms in certain long range deformations of the spin chains.

We treated the models where the sites were carrying the defining representations of the groups $SU(2)$ and $SU(3)$. The extension to other Lie groups and other representations seems rather straightforward, as long as their is an original local Hamiltonian and a local boost operator. Similarly, models with quantum group symmetries are also easily accommodated. For example the long range deformations of the XXZ chain were treated in detail in [37], and there are no essential complications as compared to the XXX case. Thus, quite generally we expect that the currents are always of the form (40), with the Gaudin matrix and the one-particle eigenvalues reflecting the given (possibly nested) Bethe Ansatz solution.

However, we do not claim that our derivation constitutes a rigorous proof. Some properties of the long range deformations, such as the convergence of the perturbation series, the quasi-locality of the resulting charges, and the effects of the truncation on the finite volume spectrum are not rigorously proven. Regarding the dilatation operator in $AdS_5/CFT_4$ the exact spectrum is well understood and completely under control (see [38,39] and references therein). On the other hand, there are fewer exact results for the general construction presented in [9], and as far as we know it is not known how to obtain a finite volume integrable model (including all corrections in $\kappa$) for an arbitrary long-range deformation. Nevertheless, our computations should be trusted as long as the perturbing operators are not longer than the length of the spin chain, and we gave a few additional arguments for this below (83).

Having obtained such simple results for the current mean values, the following questions naturally emerge: What else can we compute with similar simple tricks? And why do these results exist? The first question is easier, and here we list a few ideas for further research:

- **Connection to the general theory of short range correlators in the fundamental $SU(N)$-symmetric models.** It is known that in the XXZ and XXX Heisenberg spin chains the mean values of all local operators factorize: they can be expressed as combinations of only a few functions, which originate in the two-site density matrix of an inhomogeneous chain [40–43]. Our previous work [6,44] showed that for the $SU(2)$-invariant operators of the homogeneous XXX chain these building blocks are identical to the mean values of the currents $J_{\alpha,\beta}^0$. One of the most interesting questions is how much of this can be generalized to the higher rank cases.

  The existing results in the literature imply, that in the $SU(3)$ chain there is no factorization procedure that would express the mean values of *all* short range operators using a finite set of functions [45, 46]. Nevertheless, this does not exclude the existence of factorized formulas for some special sets of local operators. Here we derived the mean values for the infinite family of current operators in the $SU(3)$ chain, and this points to the possibility of finding also further factorized correlators.

- **Models without $U(1)$ symmetry.** The XYZ model lacks $U(1)$ symmetry, and there is no particle-like interpretation of the eigenstates in finite volume. On the other hand, the exact finite volume spectrum is in principle known [47], and the additional local charges and currents can be found in the same way with the boost operator [26]. If some generalization of the asymptotic Bethe Ansatz is found, then computing the current mean values could ultimately lead to the Generalized Hydrodynamics formulated for this model, in spite of not having a particle-like interpretation on the fundamental level.

- **Boundary spin chains.** The long-range deformations of open chains were already treated in the works [37,48], where the asymptotic Bethe Ansatz was already derived. The application of our simple arguments could lead to closed form and compact results

for the mean values of certain operators localized at the boundary. This would be interesting, because even though in principle *all* boundary correlators of the XXZ model are known in the form of multiple integrals [49,50], factorization of these integrals has not yet been observed in the boundary case.

- **Continuum models.** In principle the continuum integrable models could also be deformed, yielding the current mean values and possibly some other observables. In general we expect that the deformed Hamiltonian and charges would include an infinite series of higher derivatives, which could be perhaps summed up to some non-local, but well defined operators. Such models already appeared in the literature, although in disguise: for example the work [51] treated a generalization of the Lieb-Liniger model, where the scattering factor was modified in a non-trivial way. However, it was also found in [51] that in a proper parametrization the scattering factor has the same form as in the Lieb-Liniger model, and only the momentum is deformed, see eqs. (38) and (47) there. Clearly, this case also belongs to the class of the boost deformed models.

Let us now return to the question: why do such results exist? Regarding the current mean values it was already explained in [6], that the ultimate reason for the simple result and the exact quantum-classical correspondence is the two-particle reducibility of the Bethe wave function. On the other hand, the observed connection between correlation functions and long-range deformations still seems surprising. We believe that the present understanding of the long-range deformed integrable models is not satisfying, and deserves further research.

In closing we also note that exact results similar to ours for mean values in certain nested spin chains were derived recently using completely different methods in [52].

# Acknowledgments

The author is thankful to Yunfeng Jiang, Amit Sever and Artur Hutsalyuk for very useful discussions. Furthermore, we are once more grateful to Bruno Bertini and Lorenzo Piroli for motivating us to investigate this problem, and we also benefited from early discussions with Márton Kormos, Lorenzo Piroli and Eric Vernier. This research was supported by the BME-Nanotechnology FIKP grant (BME FIKP-NAT), by the National Research Development and Innovation Office (NKFIH) (K-2016 grant no. 119204), by the János Bolyai Research Scholarship of the Hungarian Academy of Sciences, and by the ÚNKP-19-4 New National Excellence Program of the Ministry for Innovation and Technology.

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
