# Peer review of "Current operators in integrable spin chains: lessons from long range deformations"

_SciPost Physics, doi:SciPost Phys. 8, 016 (2020)_

## Round 2 · Referee Report · Anonymous (Referee 1) · 2020-1-18

Report

This paper is concerned with the computation of exact expectation (mean) values in all excited states of current operators, which is an area of active research. For the rank-1 case, the result is already known. This paper makes an interesting and simple connection to long-range deformations of local spin chains, which allows one to recover the rank-1 result, and obtain a new result for rank 2. Various interesting generalizations now seem within reach.

---

## Round 2 · Referee Report · Anonymous (Referee 2) · 2020-1-19

Strengths

1. very clear paper
2. important extension of earlier results
3. makes a connection with earlier results that went somewhat unnoticed outside specialised literature
4. potential for impact across scientific fields (within theoretical physics)

Weaknesses

1. mostly builds on existing results (including some by the author)

Report

This article is concerned with the computation of current mean values in integrable spin chains. These integrable theories have a wealth of commuting conserved charges. For each such charge, a current can be defined in such a way to satisfy a discretised version of the continuity relation which one would encounter in a 1+1 dimensional model. Firstly, the author reviews this construction and highlights how it is not unique. Next, he turns to the problem of computing the expectation value of such current. This is something that was already considered by the author and collaborators in an earlier work using a form factor expansion. Here the author uses the Bethe wavefunction of the integrable model. To do so, he exploits a relation between these currents and the Bethe equations which was first derived in a rather different context (related to AdS/CFT integrability) by Bargheer, Beisert and Loebbert. Finally, the author applies these ideas to the SU(2) and SU(3) nearest-neighbour chains.

This article is very clearly written. Though some of the ideas here were already discussed in previous work by the author, the Bethe-ansatz approach that he employs here is more powerful than what was previously known in the literature. This is nicely illustrated by the example of the SU(3) chain, which can be rather easily dealt with by these techniques. This work is also noteworthy in that --- together with the earlier work by Bargheer et al. --- it provide a basis to study current-current deformations in lattice model. Such deformations are a topic of great interest in the high-energy theory, though the attention there has been mostly been devoted to 1+1 dimensional models.

It would be interesting to check that the author's construction does indeed extend to more general Lie algebras and representations (as the author claims). Moreover, it would be especially interesting to investigate whether these ideas can be used when the nested Bethe ansatz fails, but the model can nonetheless be studied exactly, for instance by functional Bethe ansatz techniques. I regard these as important directions for future work, but I do not think that the author should necessarily address them in this very paper. The article is clear and significant in its current form, and to my mind it is suitable for publication without any need for revision.

Requested changes

no changes are required.

---

## Editorial Decision

published